# Activation of Autophagic Flux Maintains Mitochondrial Homeostasis during Cardiac Ischemia/Reperfusion Injury

**DOI:** 10.3390/cells11132111

**Published:** 2022-07-04

**Authors:** Lihao He, Yuxin Chu, Jing Yang, Jin He, Yutao Hua, Yunxi Chen, Gloria Benavides, Glenn C. Rowe, Lufang Zhou, Scott Ballinger, Victor Darley-Usmar, Martin E. Young, Sumanth D. Prabhu, Palaniappan Sethu, Yingling Zhou, Cheng Zhang, Min Xie

**Affiliations:** 1Department of Medicine, Division of Cardiovascular Disease, University of Alabama at Birmingham, Birmingham, AL 35233, USA; lihaohe@uabmc.edu (L.H.); ychu@uabmc.edu (Y.C.); jyang@ctbu.edu.cn (J.Y.); jinhe@uabmc.edu (J.H.); yutaohua@uab.edu (Y.H.); cici3202@uab.edu (Y.C.); glennrowe@uabmc.edu (G.C.R.); lufangzhou@uabmc.edu (L.Z.); martinyoung@uabmc.edu (M.E.Y.); prabhu@wustl.edu (S.D.P.); psethu@uabmc.edu (P.S.); 2Department of Cardiology, Guangdong Provincial People’s Hospital, Affiliated with South China University of Technology, Guangzhou 510080, China; syzhouyingling@scut.edu.cn; 3The Key Laboratory of Cardiovascular Remodeling and Function Research, Chinese Ministry of Education, Chinese National Health Commission and Chinese Academy of Medical Sciences, The State and Shandong Province Joint Key Laboratory of Translational Cardiovascular Medicine, Department of Cardiology, Qilu Hospital of Shandong University, 107 Wenhuaxi Road, Jinan 250012, China; zhangc@sdu.edu.cn; 4Department of Pathology, Division of Molecular and Cellular Pathology, University of Alabama at Birmingham, Birmingham, AL 35233, USA; gloriabenavides@uabmc.edu (G.B.); scottballinger@uabmc.edu (S.B.); vdarleyusmar@uabmc.edu (V.D.-U.); 5Department of Medicine, Division of Cardiology, Washington University School of Medicine, St. Louis, MO 63110, USA

**Keywords:** autophagy, mitochondrial biogenesis, mitochondrial dynamics, ROS, myocardial ischemia/reperfusion injury

## Abstract

Reperfusion injury after extended ischemia accounts for approximately 50% of myocardial infarct size, and there is no standard therapy. HDAC inhibition reduces infarct size and enhances cardiomyocyte autophagy and PGC1α-mediated mitochondrial biogenesis when administered at the time of reperfusion. Furthermore, a specific autophagy-inducing peptide, Tat-Beclin 1 (TB), reduces infarct size when administered at the time of reperfusion. However, since SAHA affects multiple pathways in addition to inducing autophagy, whether autophagic flux induced by TB maintains mitochondrial homeostasis during ischemia/reperfusion (I/R) injury is unknown. We tested whether the augmentation of autophagic flux by TB has cardioprotection by preserving mitochondrial homeostasis both in vitro and in vivo. Wild-type mice were randomized into two groups: Tat-Scrambled (TS) peptide as the control and TB as the experimental group. Mice were subjected to I/R surgery (45 min coronary ligation, 24 h reperfusion). Autophagic flux, mitochondrial DNA (mtDNA), mitochondrial morphology, and mitochondrial dynamic genes were assayed. Cultured neonatal rat ventricular myocytes (NRVMs) were treated with a simulated I/R injury to verify cardiomyocyte specificity. The essential autophagy gene, ATG7, conditional cardiomyocyte-specific knockout (ATG7 cKO) mice, and isolated adult mouse ventricular myocytes (AMVMs) were used to evaluate the dependency of autophagy in adult cardiomyocytes. In NRVMs subjected to I/R, TB increased autophagic flux, mtDNA content, mitochondrial function, reduced reactive oxygen species (ROS), and mtDNA damage. Similarly, in the infarct border zone of the mouse heart, TB induced autophagy, increased mitochondrial size and mtDNA content, and promoted the expression of PGC1α and mitochondrial dynamic genes. Conversely, loss of ATG7 in AMVMs and in the myocardium of ATG7 cKO mice abolished the beneficial effects of TB on mitochondrial homeostasis. Thus, autophagic flux is a sufficient and essential process to mitigate myocardial reperfusion injury by maintaining mitochondrial homeostasis and partly by inducing PGC1α-mediated mitochondrial biogenesis.

## 1. Introduction

After acute myocardial infarction (MI), early reperfusion is the key to reducing infarct size [1]. However, the reperfusion part of the ischemia/reperfusion (I/R) injury is also a significant part of deciding myocardial infarct size, accounting for around 50% of the final infarct size [1]. Many pathophysiologic mechanisms lead to myocardial I/R injury, such as ion accumulation, free radical formation/reactive oxygen species (ROS), endothelial dysfunction, inflammation, and immune activation [2]. To mitigate reperfusion injury, many clinical trials were conducted to evaluate the effectiveness of therapies such as ischemia remote conditioning, adenosine, and cyclosporine infusion [3]. Some have shown beneficial effects in small clinical trials, but all failed to improve clinical outcomes in subsequent large clinical trials [3,4]. Thus, for now, there is no standard therapy to mitigate reperfusion injury [5]. Clinically appropriate and effective treatment is needed to minimize reperfusion injury, and a detailed understanding of the molecular biological pathways of reperfusion injury is vital for developing therapies.

Previous animal studies have shown that histone deacetylases (HDAC) inhibitors have strong cardioprotection in mouse models of I/R injury. Notably, we reported that an FDA-approved anticancer HDAC inhibitor, suberoylanilide hydroxamic acid (SAHA, Vorinostat (Zolinza^®^, Merck, Rahway, NJ, USA)), reduces infarct size by 40% and blunts I/R injury by maintaining autophagic flux when given at the time of reperfusion in rabbits [6]. Our following study demonstrated that the protective benefits of SAHA are not only dependent on autophagy but also on increasing mitochondrial biogenesis, which is mediated by peroxisome proliferator-activated receptor gamma coactivator-1 alpha (PGC1α) [7]. However, as an anticancer medication, HDAC inhibition has pleiotropic effects; some are unwanted [8]. We have tested a specific autophagy-inducing peptide, Tat-Beclin 1 (TB) [9], in a mouse cardiac I/R model. The only known cellular effect of TB is inducing autophagy, which allows us to study the effect of autophagy on cardiac I/R injury. We found that TB reduces infarct size when given at the time of reperfusion and reduces ROS formation [10].

Two major processes tightly regulate mitochondrial homeostasis: degradation of damaged mitochondria by autophagy (mitophagy) and mitochondrial biogenesis [11]. Our prior study showed that SAHA induces both autophagic flux and mitochondrial biogenesis mediated by PGC1α [7]. SAHA affects multiple pathways by enhancing acetylation of histones (gene expression) and acetylation of signaling proteins (post-translational regulation) [12]. For example, recent studies have shown that HDAC inhibition affects the mTOR signaling pathway [10,13] and affects the fatty acid metabolism by inhibiting succinate dehydrogenase [14]. Thus, it is possible that SAHA induces PGC1α through these pathways, not only autophagy. Whether autophagic flux induced by TB without affecting the mTOR or cardiac metabolism maintains mitochondrial homeostasis is unknown. This study aimed to test the effects of TB-induced autophagic flux on cardiomyocyte mitochondrial biogenesis and homeostasis during I/R injury in vitro and in vivo. We showed that TB-induced autophagic flux reduces ROS and increases mitochondrial mtDNA level and function measured by oxygen consumption rate (OCR) using Seahorse, PGC1α expression, and mitochondrial dynamic genes during I/R injury, which depends on ATG7 in adult mouse cardiomyocytes.

## 2. Methods and Materials

### 2.1. Experimental Animals

All animal protocols were approved by the Animal Care and Use Committee of the University of Alabama at Birmingham. All mice used in this study were male with free access to feeding and water. They were all housed in a pathogen-free environment, with a standard light/dark cycle of 12:12 h.

### 2.2. Generation of Inducible Cardiomyocyte-Specific ATG7 Knockout Mice

The inducible cardiomyocyte-specific ATG7 knockout mice were generated as previously described [7]. Briefly, the ATG7F/F mouse was obtained from Dr. Massaki Komatsu [15], and the αMHC-merCremer mouse was purchased from Jackson’s lab [16]. Tamoxifen was administered IP at 20 mg/kg 1 week before the experiment for 5 days to 8–12-week-old mice [17]. With the tamoxifen injection, the inducible myocardium-specific ATG7 knockout mice, αMHC-merCremer+; ATG7F/F (ATG7 cKO (KO)), were generated. ATG7F/F treated with tamoxifen injection (WT) were used as control. Within 1 week after the last injection, inducible cardiomyocyte-specific ATG7 knockout (KO) and wild-type (WT) littermate control mice were subjected to I/R surgeries or isolation of AMVMs.

### 2.3. Mouse I/R Model

Eight- to twelve-week-old male C57BL/6J wild-type mice or ATG7 cKO or WT littermate control mice were used for mouse I/R surgeries. Briefly, mice were anesthetized with isoflurane and placed in a supine position on an electric heating pad (37 °C). Animals were intubated and ventilated. Following the left thoracotomy, the left anterior descending coronary artery was visualized under a microscope and ligated with a reversible knot, and regional ischemia was confirmed visually. Then the ligation was released after 45 min of ischemia. Please refer to the detailed protocol in prior publications [6,7]. Specific effects of the TB were controlled by the control peptide, Tat-Scrambled (TS) [9]. For the control group, mice received TS 20 mg/kg intraperitoneal (IP) injection at the time of reperfusion [10]. For the TB experiment group, mice were given TB 20 mg/kg IP injection at reperfusion. The heart was treated with 45 min ischemia and 24 h of reperfusion. Then, the mice were euthanized, and, using a dissecting microscope, the heart was cut into three tissue zones: infarct zone (I), border zone (B), and remote zone (R). Some pieces of the heart were used for electron microscopy. Then, the samples were homogenized and divided for subsequent Western blot, DNA, and RNA isolation for real-time qPCR and qRT-PCR analyses.

### 2.4. Primary Culture of Neonatal Rat Ventricular Myocytes

Ventricles from 1- to 2-day-old Sprague-Dawley rats with mixed-sex were collected and digested using a neonatal mouse/rat cardiomyocyte isolation system (Neomyt Kit, NC6031, Cellutron, Baltimore, MD, USA). The resulting cell suspension was preplated to clear fibroblasts. These neonatal rat ventricular myocyte (NRVMs) cells were plated and cultured for 24 h in DMEM/M199 (3:1 ratio) containing 5% fetal bovine serum and 100 μmol/L bromodeoxyuridine (Sigma Aldrich, St. Louis, MO, USA) to prevent noncardiomyocyte growth, and then in serum-free culture media 2–3 days for treatment and subsequent harvest. Typical cultures were notable for >95% cardiomyocytes.

### 2.5. Adult Mouse Cardiomyocyte (AMVM) Isolation

The AMVM isolation was performed as previously described [18] with slight modifications using a Langendorff perfusion system. Briefly, the heart of each male mouse was cannulated through the aorta and perfused with perfusion buffer (in mM: 113 NaCl, 4.7 KCl, 0.6 KH_2_PO_4_, 0.6 Na_2_HPO_4_, 1.2 MgSO_4_, 12 NaHCO_3_, 10 KHCO_3_, 10 HEPES, 30 taurine, 10 BDM, and 5.5 glucose) after isolating from the adult mouse, followed by digestion buffer (perfusion buffer supplemented with 300 U/mL collagenase II and 1.5 mg/mL proteinase XIV). Then, the heart was minced and filtered through a 100 µm-cell strainer. The concentration of Ca^2+^ was gradually increased to a final concentration of 900 µM with repeated centrifugation and resuspension. Finally, the cells were resuspended and plated on laminin-coated tissue culture dishes in the plating medium (DMEM 4.5 g/L glucose, supplemented with FBS, blebbistatin, and penicillin/streptomycin) at 37 °C and 5% CO_2_ for 1–3 hours. Then the cells were cultured with DMEM 4.5 g/L glucose supplemented with BSA and blebbistatin. Cell purity was greater than 98% of cardiomyocytes.

### 2.6. Simulated I/R in Cultured Cells

Simulated I/R of cardiomyocytes was performed as described before [6,7]. Briefly, ischemia was performed using an ischemia-mimetic solution and a humidified gas chamber with 95% N_2_ and 5% CO_2_. After 2 h of simulated ischemia, reperfusion was initiated by changing to a normoxic NRVM culture medium with 5% horse serum and incubation in an incubator with 95% room air, 5% CO_2_ [19]. Controls incubated in a normoxic NRVM culture medium with 5% horse serum were prepared in parallel for each condition. An amount of 2.5 μmol/L TB or TS was added at reperfusion in acidified Opti-MEM media [9] for a total of 4 to 6 h of treatment in the group in which cells were treated with sI/R of 2 h ischemia and 4 to 6 h of reperfusion. LC3 Western blots with bafilomycin A were performed to assay autophagic flux. For the AMVM, the ischemia time was 1 h, and the reperfusion time was 2–4 h.

### 2.7. Reagents

GAPDH (10R-G109a, Fitzgerald, Acton, MA, USA), ATG7 (#2631, Cell Signaling Technology, Danvers, MA, USA), Fis1 (# PA5-22142, Invitrogen, Thermo Fisher Scientific, Waltham, MA, USA), Drp1 (ab184247, Abcam, Waltham, MA, USA), Mfn1 (sc-166644, Santa Cruz Biotechnology, Dallas, TX, USA), and Mfn2 (ab50843, Abcam, Waltham, MA, USA) antibodies were purchased. Rabbit anti-LC3 polyclonal antibody was given to us as a generous gift from Dr. Hill’s lab (UT Southwestern Medical Center, Dallas, TX, USA). TS and TB were purchased from Sigma (customized Tat-Beclin 1, YGRKKRRQRRRGGTNVFNATFEIWHDGEFGT, Tat-Scrambled, YGRKKRRQRRRGGVGNDFFINHETTGFATEW). Bafilomycin A was purchased from LC laboratories (B-1080, LC Laboratories, Woburn, MA, USA).

### 2.8. Fluorescent Microscopy

Dichlorofluorescein (H2DCFDA, MP36103, 10 μM, Invitrogen, Thermo Fisher Scientific, Waltham, MA, USA), MitoSOX Red (M36008, 5 μM, Invitrogen, Thermo Fisher Scientific, Waltham, MA, USA), and tetramethylrhodamine methylester (TMRM, I34361, T668, 0.1 μM, Invitrogen, Thermo Fisher Scientific, Waltham, MA, USA) were used to measure cellular ROS levels, mitochondrial ROS levels, and mitochondrial membrane potential, respectively. As previously described [7], NRVMs were incubated in H2DCFDA for 30 min, in MitoSOX for 10 min, and in TMRM for 30 min. Images were immediately captured by using a Nikon Eclipse Ti fluorescent microscope. We used the ImageJ software (1.53, NIH, Bethesda, MD, USA) to determine the average fluorescence intensity of H2DCFDA, MitoSOX, and TMRM in each group.

### 2.9. Isolation of DNA and RNA, Measurement of mtDNA Copy Number, and mtDNA Damage

Following the previous protocols [7], total DNA and RNA were isolated from NRVMs, AMVMs, and the three zones of the mouse hearts. Mitochondrial DNA (mtDNA) content, intact mtDNA (16.2 kb), and total mtDNA (0.22 kb) were measured by quantitative PCR (qPCR) and semiquantitative PCR as described before [7,20]. Data were expressed as the ratio of the intact mtDNA divided by the total mtDNA for each group. A decreased intact/total ratio is indicative of increased mtDNA damage. ImageQuant (V7, GE Healthcare, Chicago, IL, USA) was used to quantify the density of semiquantitative PCR products. Mitochondrial DNA-specific gene ATP6 or COXII primers were used to perform the qRT-PCR.

### 2.10. Assays of Cell Death

NRVMs cell death was detected by using an MTT Assay Kit (V13154, 5 mg/mL, Invitrogen, Thermo Fisher Scientific, Waltham, MA, USA). NRVMs were treated with sI/R of 5 h ischemia and 3 h of reperfusion, then changed to phenol red-free culture medium (in mM: 130 NaCl, 4 KCl, 1.25 MgSO_4_, 1.2 CaCl_2_, 6.25 NaHCO_3_, 20 HEPES, 20 D glucose) with 10% MTT solution. Incubated at 37 °C overnight, we then added SDS-HCl solution (SDS 0.1 g/mL, 0.01 M HCl), the same volume of the culture medium to mix. Incubated at 37 °C for 4 h, then pipetted well to mix sample again, and read the absorbance at 570 nm. AMVMs were treated with sI/R of 1 h of ischemia and 4 h of reperfusion. A Nikon Eclipse Ti fluorescent microscope was used to take images at the same magnification in the bright field before sI/R and after 4 h of reperfusion. Live cell numbers were counted by using ImageJ software (1.53, NIH, Bethesda, MD, USA).

### 2.11. Electron Microscope (EM)

EM was performed as described before [7]. A Tecnai Spirit T12 transmission electron microscope, operating at 20 to 120 kV and equipped with a digital camera, was used to image the heart sections. Mitochondria numbers and size were analyzed by using ImageJ software (1.53, NIH, Bethesda, MD, USA). The EM image reader was blinded for group assignment, and the mitochondrial number and size were an average of 4–6 images per mouse.

### 2.12. ATP Concentration Measurement in AMVMs

The ATP levels in AMVMs were detected by using an ATP Determination Kit (A22066, Invitrogen, Thermo Fisher Scientific, Waltham, MA, USA). AMVMs were treated with sI/R of 1 h of ischemia and 2 h of reperfusion. Boiling water was used to inhibit ATPase [21]. AMVMs suspension was collected and centrifuged at 12,000× *g* for 5 min at 4 °C. The supernatant was used for bioluminescence measurement and read at 560 nm. The standard curve of ATP was obtained by serial dilutions of 10 nM ATP solution.

### 2.13. Cellular Bioenergetics

Analyses of cellular bioenergetics were performed using the Seahorse XFe96 Extracellular Flux Analyzer (Agilent, Santa Clara, CA, USA) [22,23]. After isolation, the NRVMs were plated into an Agilent Seahorse XF96 cell culture microplate (Product No.101085-004) at 28,000 cells/well for 4 days. On the day of the experiment, the cells were exposed to 2 h of ischemia as described above, then 2.5 µM TS/TB was added for a 4-hour reperfusion. After IR or normoxia, the cells were changed into the XF media (nonbuffered DMEM supplemented with 5.5 mM glucose, 1 mM pyruvate, and 4 mM glutamine, pH 7.36) at 37 °C. Oxygen consumption rate (OCR) was measured for basal OCR (OCR before oligomycin minus OCR after antimycin) followed by sequential injections of 1 µg/mL oligomycin, 1 µM FCCP, and 10 µM antimycin A. Mitochondrial parameters were calculated for ATP-linked (OCR before oligomycin minus OCR after oligomycin), proton leak (OCR after oligomycin minus OCR after antimycin), maximal (OCR after FCCP minus OCR after antimycin), reserve capacity (OCR after FCCP minus OCR before oligomycin), and nonmitochondrial (OCR after antimycin).

### 2.14. Statistical Methods

Averaged data are reported as mean ± standard error of the mean. Data were analyzed with the unpaired Student *t* test for 2 independent groups, paired *t* test for dependent data, and the 1-way or 2-way analysis of variance followed by the Tukey’s post hoc test for pairwise comparisons. Studies with repeat measures were analyzed using repeated-measures analysis of variance. For all statistical tests, a *p* value of <0.05 was considered statistically significant, and all tests were 2-tailed. All statistical analyses were performed using GraphPad Prism (version 8.0.2, GraphPad Software, San Diego, CA, USA) software. Detailed statistical methods that were used for each set of data are listed in the online supplemental.

## 3. Results

### 3.1. Tat-Beclin 1 Induces Autophagy and Reduces Oxidative Stress in Neonatal Rat Ventricular Cardiomyocytes (NRVMs) Simulated I/R Injury

Autophagy is a conserved intracellular process to recycle cellular contents, including damaged organelles [24]. Previous studies demonstrated that the activation of autophagy during reperfusion could reduce infarct size [25,26]. TB has an 18 amino acid region of the essential autophagy gene Beclin-1 that can induce autophagy by releasing native Beclin-1 from its inhibitor GAPR-1 to promote autophagy [9]. The only known function of TB is to induce autophagy. Autophagy is a dynamic process of flux. So, an increased steady-state level of autophagosome or autophagosome-related proteins means either an increase in autophagic flux, or a block in downstream lysosomal processing of these autophagosomes, or both [24]. To measure the autophagic flux, we used bafilomycin A as a lysosomal activity blocker to block the downstream processing. Autophagy was evaluated by Western blot detection of the autophagosome-associated lipidated isoform of LC3 (LC3-II). NRVMs were exposed to sI/R (2 h of ischemia and 4 h of reperfusion) with the treatment of TS/TB at the reperfusion time. In the TS group, steady-state levels of LC3-II were slightly decreased after I/R injury. Reduced levels of LC3-II demonstrated a decline in autophagic flux in the TS group observed after bafilomycin A treatment. With the absence of bafilomycin A, LC3-II levels were similar in both TS and TB treatment groups. However, blocking lysosomal activity by bafilomycin A revealed a significant increase in autophagic flux induced by TB, as proved by the accumulation of LC3-II (Figure 1A,B).

Many studies have uncovered the role of ROS in cardiac I/R injury, which shows ROS is an important contributor to I/R injury, especially mitochondrial ROS [27]. Limiting the generation of ROS can reduce infarct size and cardiac I/R injury [28]. Therefore, we examined the ROS levels in NRVMs subjected to I/R (sI/R, 2 h of ischemia and 4 h of reperfusion) injury with TS or TB treatments at the time of reperfusion. We used H2DCFDA and MitoSOX staining to quantify cellular and mitochondrial ROS levels in cardiomyocytes by microscopy. Cytosolic ROS levels after I/R were significantly increased in the TS-treated control group, while the ROS levels were reduced significantly in the TB-treated group (Figure 1C,D). Similar results were observed in the mitochondrial ROS levels measured by mitoSOX (Figure 1E,F). In aggregate, these data indicate that during I/R injury, TB treatment in NRVMs increases autophagic flux and suppresses I/R-induced damaging cytosolic ROS levels.

### 3.2. Tat-Beclin 1 Treatment Increases Mitochondrial DNA Levels, Partially Maintains Mitochondrial Function, and Reduces Cell Death in Cardiomyocytes Subjected to Simulated I/R Injury

Mitochondria, the major source of ROS and the major antioxidant producers, play a critical role in many cell mediating processes [29]. Moreover, mitochondria are the major targets for ROS damage [27]. Damaged mitochondria will initiate cell death pathways, contributing to cardiomyocyte death and heart failure [30]. Excessive generation of ROS during reperfusion injury damages mitochondrial DNA (mtDNA), protein, and lipids, which leads to more ROS production and, as a vicious circle, causes mitochondrial dysfunction [31]. Thus, mitochondrial homeostasis is a critical target to mitigate I/R injury by maintaining energy production and reducing ROS levels. We have shown that TB reduced ROS levels in NRVMs. We next analyzed whether TB would affect mtDNA levels during I/R injury. We measured the intact and total mtDNA (16.2 and 0.22 kb, respectively) using semi-qPCR to evaluate mtDNA levels in NRVMs [20]. The ratio of intact mtDNA to total mtDNA serves as an indicator of mtDNA damage. We used 6 h of reperfusion to show the later effects on mitochondrial DNA. During I/R, the intact/total mtDNA ratio in TS treatment group cardiomyocytes decreased, indicating mtDNA damage. Compared with the TS-treated NRVMs subject to I/R injury, the intact/total mtDNA ratio in the TB treatment group was significantly higher (Figure 2A,B). For the total mtDNA levels, the mtDNA copy numbers of ATP6 and COX-II during I/R in NRVMs were significantly increased by with the treatment of TB compared with the TS group. In the normoxia group, there was a trend of increasing mtDNA with the TB treatment (Figure 2C,D).

Mitochondrial membrane potential plays a crucial role in mitochondrial function. During I/R, cellular calcium increases, and subsequent activation of caspases leads to the failure of mitochondrial membrane potential and results in cell death [30]. We next measured the mitochondrial membrane potential of NRVMs by using TMRM staining and visualized by fluorescence microscopy. We found a significant decrease in TMRM fluorescence after I/R in the TS group. By contrast, TB treatment preserved mitochondrial membrane potential after I/R (Figure 2E,F).

Mitochondrial oxidative phosphorylation is required to generate ATP and restore calcium gradients following cardiac ischemia–reperfusion [32]. We found that TB maintains mitochondrial membrane potential after I/R, similar to SAHA [7]. Recent advances in measuring bioenergetics using Seahorse have opened the door to measuring cardiac mitochondrial function [33]. We optimized the protocol to measure oxygen consumption rate (OCR) in cardiomyocytes. Then, we measured the cellular bioenergetics by measuring the OCR in NRVMs and the cellular ATP concentration of AMVMs subjected to I/R injury. There was no significant difference between the OCR of the TS- and TB-treated group in the normoxia treatment. However, after I/R, the maximal and the reserve capacity OCR was lower in the TS-treated group, and TB partially preserved these parameters (Figure 2K,L). There was no difference between the TS and TB groups after I/R in other OCR measurements, including basal, ATP linked, and proton leak (Figure 2H–J). This indicates that TB treatment partially protects cellular bioenergetic function post-I/R injury. Of note, this is different from the effect of HDAC inhibition on OCR in cardiomyocytes during I/R injury, which reduces the OCR and also affects fatty acid metabolism [14]. Due to the limited number of AMVMs, we measured the ATP concentration in AMVMs subjected to I/R. TB increased ATP levels in both the normoxia and I/R treatment groups (Appendix A), suggesting that TB has similar protective effects in AMVMs. As mitochondria also play an important part in several types of cell death [29], we measured cardiomyocyte viability after I/R. Before and after I1R4, viable AMVMs were counted. TB reduced AMVMs death significantly after I/R (Appendix A). Similar to our published data [10], the MTT assay in NRVMs showed that the TB group had higher viability than the TS group after I/R injury (Appendix A). In summary, TB partially preserves intact functional mtDNA, maintains cardiomyocytes’ bioenergetics, and reduces cell death after I/R.

### 3.3. Tat-Beclin 1 Treatment Induces Autophagy and Maintains Mitochondrial Homeostasis in Mouse Myocardium Treated with I/R Injury

We established the beneficial effects of TB on autophagy and mtDNA during I/R in vitro. To further explore the effects of TB in vivo, wild-type C57BL/6J mice were subjected to cardiac I/R injury, and TS/TB was given at the time of reperfusion. After 45 min of ischemia and 24 h of reperfusion, mice were euthanized, and the hearts were isolated and divided into three tissue zones: infarct zone (I), border zone (B), and remote zone (R) as in the previous studies [6,7]. The remote zone was considered myocardium without ischemia. In the border zone, LC3-II levels in the TB-treated group increased significantly more than in the TS-treated group. In the ischemic and remote zones, the LC3-II levels in the TB-treated groups were also higher but did not reach statistical significance (Figure 3A,B). Similarly, mtDNA content detected by real-time qPCR using primers specific to ATP6 and COX-II in the border zone was also significantly increased in the TB group compared with the TS group (Figure 3C,D). In summary, TB induced an increase in LC3-II after I/R in the mouse heart border zone, suggesting increased autophagic flux, and the increased autophagic flux preserved mtDNA content.

Next, we evaluated the effect of TB on mitochondrial biogenesis and fission- and fusion-related gene expression in the mouse heart subjected to I/R injury. PGC1α gene expression in the border zone of the TB group was significantly increased compared to the TS group, indicating possible increased PGC1α-mediated mitochondrial biogenesis (Figure 3E). Since mitochondrial homeostasis is a dynamic process involving mitochondrial fission and fusion [34], we then measured the expression of mitochondrial dynamic genes. In the infarct border zone of the mouse heart with I/R injury, the expression of mitochondrial fission gene, Drp1 (Figure 3F), and fusion genes Opa1, Mfn1, and Mfn2 (Figure 3H–J) were significantly induced in the TB group than the TS group. The fission gene, Fis1, had a trend of increase but did not reach statistical significance in the TB group. These indicated that TB increased mitochondrial dynamics in the mouse heart border zone subjected to I/R injury. We had reported that SAHA treatment increases autophagosomes in rabbits and increases mitochondrial size in the mouse heart border zone using EM [6,7]. Similarly, we observed an increase in autophagosomes in the TB-treated group than in the TS group in the border zone, with autophagosomes engulfing damaged mitochondria in EM images (Appendix A). TB also increased the mitochondrial size in the mouse heart infarct border zone (Figure 3K,L). The mitochondrial sizes in the ischemic and remote zones were not significantly different (Appendix A). Collectively, these results indicate that TB treatment maintained mitochondrial homeostasis in the border zone in mouse hearts during I/R injury.

### 3.4. Tat-Beclin 1 Increases Mitochondrial Biogenesis and Dynamics, Which Is Dependent on the Essential Autophagy Gene, ATG7, in Adult Mouse Ventricular Cardiomyocytes (AMVMs)

Mitochondrial biogenesis is partially regulated by PGC1α, a central orchestrator of oxidative metabolism [35,36]. Importantly, in the kidney and brain, mitochondrial biogenesis is not increased by I/R injury, despite induction of PGC1α, suggesting removal of damaged mitochondria is needed to allow mitochondrial biogenesis. We have shown that SAHA protects hearts from I/R injury by inducing both autophagy and PGC1α-dependent mitochondrial biogenesis, and SAHA’s protective effects on mtDNA depend on PGC1α, which indicates the subsequent PGC1α-mediated mitochondrial biogenesis is needed for mitochondrial homeostasis even with increased autophagy [7]. By using NRVMs, we have also shown that SAHA’s cardiac protective effects in PGC1α-mediated mitochondrial biogenesis are cardiomyocytes specific [7]. Since the NRVMs are not mature cardiomyocytes, this time, we tested the effects of TB on PGC1α expression in AMVMs lacking ATG7.

Inducible cardiomyocyte-specific ATG7 knockout mice (αMHC-merCremer+; ATG7F/F treated with tamoxifen, ATG7 cKO (KO)) were generated as described before [7], and the loss of ATG7 was verified both by Western blots and qRT-PCR in isolated AMVMs (Figure 4A–C). The loss of ATG7 essentially abolished autophagic flux as shown in the LC3 Western blot (Figure 4A). ATG7F/F treated with tamoxifen injection were used as wild-type (WT) control. At baseline condition, TB increased PGC1α gene expression in WT cardiomyocytes up to 70% (Figure 4D). To see the effects of autophagic flux induced by TB on mitochondrial dynamics, we assessed the expressions of five genes, Drp1 and Fis1, which are associated with mitochondrial fission, and Opa1, Mfn1, and Mfn2, which are associated with fusion. Similar to the expression of PGC1α, TB treatment significantly increased the expression of Drp1, Fis1, Mfn1, and Mfn2 (Figure 5H,I). The fusion gene, Opa1, had a trend of increase but did not reach statistical significance in the WT group. Together, these data showed that TB treatment induces PGC1α expression and promotes mitochondrial dynamic gene expressions in AMVMs.

We have established that TB increases mtDNA content and increases PGC1α and mitochondrial fission and fusion genes in AMVMs. However, since TB is a peptide that may regulate other pathways rather than only inducing autophagy, we next tested whether TB’s effects on mitochondria are dependent on the crucial autophagy-related gene, ATG7. TB increased PGC1α gene expression in the WT mice group compared with TS. In ATG7 cardiomyocyte null mice, TB was not sufficient to induce the PGC1α expression (Figure 4D). Regarding the mitochondrial fission and fusion genes, TB-induced mitochondrial fission and fusion gene expressions were abolished in ATG7 cKO AMVMs (Figure 4E–I). In addition, TB increased mtDNA copy number of ATP6 was blocked by the loss of the ATG7 gene (Figure 4J). Together, these results demonstrate that TB’s ability to maintain mitochondrial homeostasis is dependent on autophagy in the AMVMs.

The detailed pathway of how TB increases PGC1α gene expression is unknown. We tested two upstream regulators of PGC1α gene, NRF2 [37] and PARIS [38], but no significant change was found (Appendix A). However, this does not exclude that these factors may still regulate PGC1α expression since these proteins are post-translationally modified, and further investigation is required. We also tested the expression levels of proteins related to mitochondrial fission and fusion. The Mfn2 level was increased, Fis1, Drp1, and Mfn1 had some trend but did not reach significance (Appendix A), also suggesting the role of post-translational modifications of these proteins.

### 3.5. The Beneficial Effects of Tat-Beclin 1 on Mitochondria in Mouse Hearts Depend on Autophagy

After we established that TB induced beneficial mitochondrial effects in AMVM, we moved on to test whether the same was true in the mouse myocardium. ATG7 cKO mice were generated, and the loss of ATG7 gene expression was verified in the myocardium (Figure 5A). The partial loss of ATG7 in the myocardium is due to the presence of other types of cells in addition to cardiomyocytes. The mitochondrial biogenesis-related gene, PGC1α, was increased in the WT mice TB group but not in the ATG7 cKO group (Figure 5B). At the same time, TB-induced Drp1 and Fis1 (mitochondrial fission genes) expression were lost in the absence of the ATG7 gene (Figure 5C,D). Similarly, TB-augmented mitochondrial fusion gene expressions were abolished in the absence of the ATG7 gene in cardiomyocytes (Figure 5E–G). Furthermore, the TB-induced mtDNA level increase measured by ATP6 gene was abolished in ATG7 cKO mice (Figure 5H). Together, these data demonstrate that in the mouse myocardium, TB maintains mitochondrial homeostasis via activating autophagy.

## 4. Discussion

In cardiac I/R injury, approximately 50% of cell death is due to reperfusion injury, for which there are no standard clinical therapies [5]. Many signaling pathways have been manipulated to mitigate reperfusion injury without success [39]. Initial appropriate mitochondrial quality control is critical for cardioprotection following I/R [3,40]. Previous studies reported that the HDAC inhibitor, SAHA, given at the time of reperfusion, blunts I/R injury by inducing cardiomyocyte autophagy and maintaining mitochondrial homeostasis [6,7]. Yet, as a nonselective HDAC inhibitor, SAHA has various untoward effects. A better therapeutic is needed to induce autophagy specifically. We previously showed that TB reduces infarct size when administered at the time of reperfusion [10]. Although SAHA induces mitochondrial biogenesis during cardiac I/R, it has other effects other than inducing autophagy, such as inhibiting the mTOR pathway and fatty acid metabolism [10,13,14]. Whether pure autophagy activation maintains mitochondrial homeostasis during I/R remains obscure. Tat-Beclin 1 was used to induce autophagic flux by binding the Beclin 1 inhibitor GARP1 specifically without affecting mTOR and fatty acid metabolism [9,10]. Here, we extended our research to test the effects of TB-induced autophagic flux on ROS, and mitochondrial function and homeostasis. With this approach, we discovered that direct activation of autophagic flux by TB can reduce I/R-induced ROS levels. Furthermore, we found that TB treatment during reperfusion preserved mtDNA and induced PGC1α mitochondrial biogenesis. Of note, the mitochondrial function measured by OCR using Seahorse was significantly preserved by TB treatment at reperfusion, which provided definitive evidence of improved mitochondrial function. We also identified the beneficial effect of TB on the induction of cardiac mitochondrial fission and fusion genes during I/R injury. Finally, we demonstrated that the benefit of mitochondrial homeostasis of TB is cardiomyocyte-specific and through bona fide activation of autophagic flux by silencing the essential autophagy gene, ATG7, in adult mouse cardiomyocytes.

### 4.1. Mitochondrial Homeostasis Is a Possible Therapeutic Target for I/R Injury

Mitochondria play an important role in regulating energy production and ROS regulation, which may be interrupted by pathological insults. Mitochondria are not only essential for meeting cellular bioenergetic needs but also contribute to several types of cell death and autophagy [29]. The natural compensation of mitochondria to these pathological stimuli includes induction of antioxidant enzymes, mitochondrial biogenesis, increased expression of respiratory complex subunits, and the metabolic shift to glycolysis [41,42]. Mitochondria may be generated in response to physiological or pathophysiological stimuli, such as oxidative stress, increased energy requirements of the cells during exercise training, electrical stimulation, changes of hormones during development, congenital mitochondrial diseases, and cardiac I/R injury [42,43,44]. Thus, mitochondrial homeostasis, including removing damaged mitochondria and generating new mitochondria, is a potential target for treating diseases lacking effective therapy, such as I/R injury in the heart and other congenital mitochondrial diseases [45,46,47].

### 4.2. Autophagic Removal of Damaged Mitochondria Becomes an Effective Mechanism to Prevent Cardiac Reperfusion Injury

Autophagy is an evolutionarily conserved process required for cellular constituent recycling (including mitochondria), which is impaired during I/R [6,48]. One study has shown that ROS overload induces mtDNA damage during cardiac reperfusion injury, and if the damaged mitochondria cannot be cleared promptly, the cell will go into apoptosis [49]. Thus, the removal of damaged mitochondria by autophagy (mitophagy) is essential for cardiomyocyte survival. We have reported previously that TB-induced autophagic flux reduced infarct size when given at the time of reperfusion [10]. We also reported that the HDAC inhibitor, SAHA, reduces infarct size by 40% when given at the time of reperfusion in rabbits through induction of autophagic flux and the promotion of mitochondrial biogenesis in mouse myocardium [6,7]. However, HDAC inhibition has broad effects and may have other unidentified off-target effects. To further explore the role of autophagy on cardiac mitochondrial homeostasis in preventing reperfusion injury, we reported here that TB peptide given at the time of myocardial reperfusion reduces ROS levels and increases mtDNA levels, likely through PGC1α-mediated mitochondrial biogenesis. Mitochondria are critical in the early responses to I/R injury. It has been reported that damaged mtDNA, opened permeability transition pore (PTP), ROS production, and decreased ATP production by mitochondria in the myocardium occur during reperfusion injury [41,50]. In cardiomyocytes, autophagy promoted mitochondrial clearance and improved mitochondrial function against oxidative stress-mediated damage [51]. We believe that the initial appropriate mitochondrial quality control and reduction in ROS production are critical for cardioprotection during reperfusion injury and provide ATP for later responses and repair [52,53]. Autophagy might be more potent than mitophagy alone in reducing reperfusion injury since it also removes other damaged organelles and protein aggregates and generates energy for repair [3]. Our data showed that inducing autophagic flux by Tat-Beclin 1 increases mitochondrial biogenesis and mitochondrial dynamics gene expressions. We speculate that augmentation of mitophagy alone may also have the same cardiomyocyte protection effects. However, the lack of a single powerful mitophagy-inducing agent prevented us from testing this [54,55,56].

### 4.3. Tat-Beclin 1-Induced Autophagic Flux Maintains Mitochondrial Homeostasis and Possibly Dynamics during Cardiac I/R Injury

Mitochondria are the primary source of ROS and also the major target for ROS damage [27]. Ischemia damages mitochondria complexes, and during reperfusion, excessive ROS will be generated through reversed electron transport at complex I [57]. In turn, ROS will further damage the mitochondria, resulting in a vicious cycle. To mitigate the I/R injury, the vicious cycle needs to be stopped [31]. One way to do it is to remove the damaged mitochondria and replace them with new, functional mitochondria. Our data showed that I/R-induced ROS production was blocked by the autophagic clearance of damaged mitochondria. After removing damaged mitochondria, mitochondrial biogenesis is needed to replete the depleted mitochondrial pool [58]. Our data showed that the mtDNA level is increased after TB treatment in both the cardiomyocytes and the mouse myocardium. These beneficial effects of TB on mitochondria are in the infarct border zone, where active cell death happens during reperfusion injury. As noted, mitochondrial homeostasis is a dynamic process; we demonstrated that TB increases mitochondrial biogenesis, possibly through increasing PGC1α-mediated biogenesis and enhancing mitochondrial fission and fusion during I/R injury. However, we only measured the fission and fusion gene and protein expression, suggesting an increase in mitochondrial dynamic. Further studies on mitochondrial fission and fusion need to be performed to verify this.

Since TB may have unintended effects in addition to inducing autophagic flux as it is designed, we used ATG7 cKO mice to test whether loss of ATG7, an essential autophagy-related protein, blocks TB’s beneficial effects. With the absence of the ATG7 gene, the beneficial effects of the TB peptide are blocked, which confirms that TB’s beneficial effect on mitochondria is through the genuine autophagy pathway. These data are consistent with our previous study showing that HDAC inhibition’s beneficial effect on mitochondria depends on ATG7 in cardiomyocytes [7]. Additionally, because overactivation of autophagy promotes pathologic cardiac remodeling [59], the therapeutic dose window of TB is narrow [10]. We should take this into consideration when using TB in a possible clinical trial.

### 4.4. The Mechanism of Autophagy Induced PGC1α-Mediated Mitochondrial Biogenesis

Besides autophagy, mitochondrial biogenesis is important in maintaining mitochondrial homeostasis during pathological stresses in I/R injury [11]. Mitochondrial biogenesis is partially regulated by PGC1α, which is essential for metabolic function and oxidative states [35,36]. Increased mitochondrial biogenesis through PGC1α after exercise contributes to improved recovery from I/R injury in the brain [60]. PGC1α can also be induced by I/R injury with ROS overproduction in the brain and kidneys [61,62]. However, PGC1α induction alone did not reduce ROS production and I/R injury in the kidneys and brain since mitochondrial biogenesis might be overwhelmed and inhibited by the harmful consequences of excessive mitochondrial damage [61,62]. Thus, PGC1α-dependent mitochondrial biogenesis may be autophagy-dependent during I/R. Consistent with this, autophagy is essential for generating signals that promote exercise-induced mitochondrial biogenesis [63]. In our study, PGC1α expression was significantly induced by TB-induced autophagic flux in cardiomyocytes. This has not been reported before and has important implications for using TB in maintaining mitochondrial homeostasis during I/R injury and even in other congenital mitochondrial diseases [45,46,47]. The loss of ATG7 prevented TB from increasing PGC1α expression, which verified that TB-induced PGC1α expression is through the autophagy pathway similar to HDAC inhibition [7]. We have tested two regulators of PGC1α expression, Nrf2 and PARIS, but did not see any significant difference [37,38]. The molecular mechanisms underlying autophagy-induced PGC1α-mediated mitochondrial biogenesis need more investigation. We speculated that there might be sensors of the mitochondria abundance level. The reduction in mitochondrial abundance by autophagic removal during I/R may trigger the expression of mitochondrial biogenesis genes, such as PGC1α. In addition, we showed that there are increases in both fission and fusion gene expression after TB treatment. This is indicative of possible increased mitochondrial dynamics. However, more definitive studies need to be performed to evaluate mitochondrial fission and fusion further.

## 5. Conclusions

Directly activated autophagic flux by a novel and powerful autophagy-inducing peptide, Tat-Beclin 1, reduces I/R-induced ROS levels and maintains mitochondrial homeostasis during cardiac I/R injury. Induction of autophagic flux also augments PGC1α expression, therefore regulating mitochondrial biogenesis in adult cardiomyocytes. Furthermore, induction of autophagic flux increased most of the mitochondrial fission and fusion gene expression, indicating possible increased mitochondrial dynamics. Additionally, these protective effects are dependent on the essential autophagy-related gene, ATG7, in adult cardiomyocytes (Figure 6). Maintaining appropriate activation of autophagic flux and mitochondrial biogenesis by Tat-Beclin 1 or other agents may be used in future myocardial protection clinical trials in ST-elevation myocardial infarction patients.

## Figures and Tables

**Figure 1 cells-11-02111-f001:**
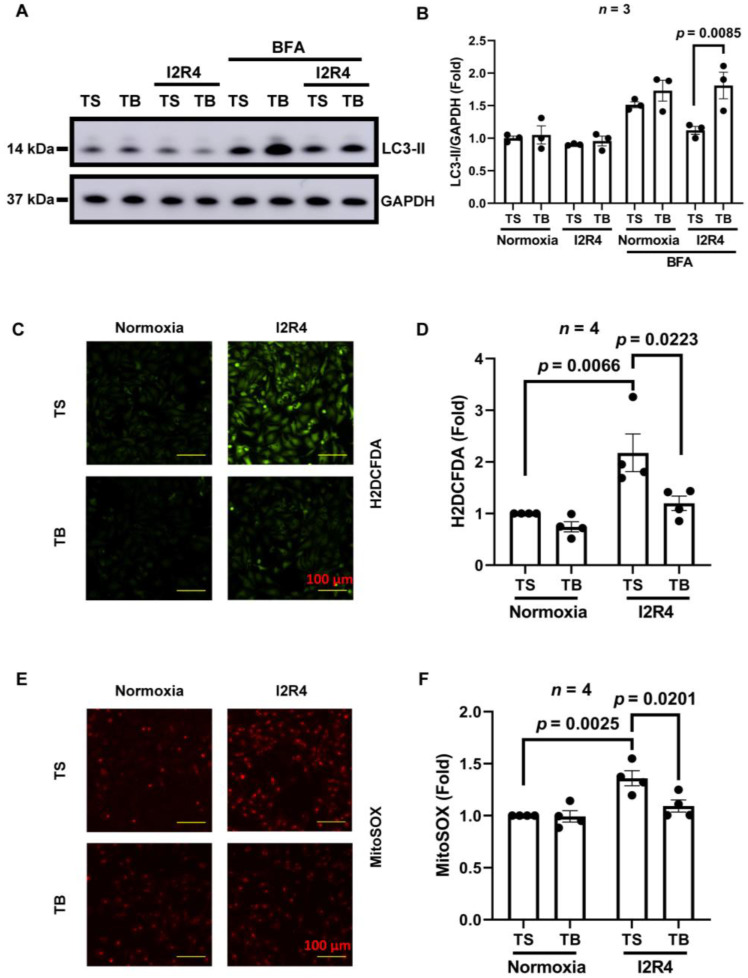
Tat-Beclin 1 induces autophagy and reduces oxidative stress in neonatal rat ventricular cardiomyocytes (NRVMs) treated with simulated I/R injury. NRVMs were treated with simulated I/R and TS or TB were added at the time of reperfusion. After 2 h of ischemia and 4 h of reperfusion, the whole-cell lysate was used for Western blot. (**A**,**B**), representative Western blot and quantification of LC3-II level in NRVMs. *n* = 3, *p* = 0.0085. (**C**,**D**), representative fluorescence microscope images of H2DCFDA staining of cytosolic ROS and quantification of fluorescence intensity in NRVMs. Bar = 100μm. *n* = 4. NC TS vs. IR TS, *p* = 0.0066. IR TS vs. IR TB, *p* = 0.0223. (**E**,**F**), representative fluorescence microscope images of MitoSOX staining of mitochondrial ROS and quantification of fluorescence intensity in NRVMs. Bar = 100 μm. *n* = 4. NC TS vs. IR TS, *p* = 0.0025. IR TS vs. IR TB, *p* = 0.0201. TS, Tat-Scrambled; TB, Tat-Beclin 1; I2R4, ischemia 2 h and reperfusion 4 h.

**Figure 2 cells-11-02111-f002:**
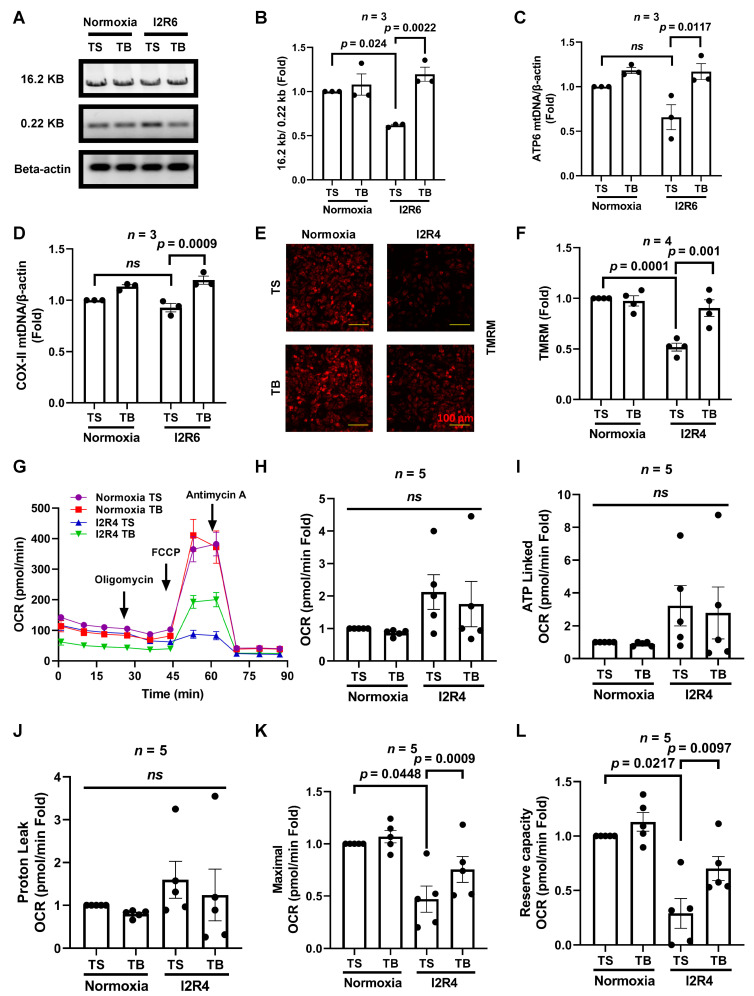
Tat-Beclin 1 increases mitochondrial DNA levels, and partially maintains mitochondrial membrane potential and cellular bioenergetics in NRVMs treated with simulated I/R injury. NRVMs were treated with TS or TB at the time of reperfusion. After simulated I/R injury of 2 h of ischemia and 4 to 6 h of reperfusion, DNA was extracted and amplified to evaluate mtDNA damage and integrity. (**A**,**B**), representative DNA gel of the intact mtDNA (16.2 kb) and total mtDNA (0.22 kb) products from semi-qPCR and quantification of the 16.2 kb/0.22 kb ratio in NRVMs. *n* = 3. Normoxia TS vs. IR TS, *p* = 0.024. IR TS vs. IR TB, *p* = 0.0022. (**C**,**D**), rat-specific ATP6 and COX-II primers for real-time qPCR were used to analyze mtDNA copy numbers in NRVMs. *n* = 3, ATP6, normoxia TS vs. IR TS, *p* = NS (not significant). IR TS vs. IR TB, *p* = 0.0117. COX-II, normoxia TS vs. IR TS, *p* = NS. IR TS vs. IR TB, *p* = 0.0009. To evaluate mitochondrial membrane potential, TMRM staining was used in NRVMs after IR. (**E**,**F**), representative images of TMRM staining and quantification of NRVMs under normoxia or I/R conditions. Bar = 100 μm. *n* = 4. Normoxia TS vs. IR TS, *p* = 0.0001. IR TS vs. IR TB, *p* = 0.001. (**G**–**L**), oxygen consumption rates for NRVMs were measured by using a Seahorse XFe96 Analyzer. (**G**). Representative OCR measurement is shown. Each parameter was calculated based on 5 independent experiments. All parameters were normalized to normoxia TS values. (**H**). Basal OCR, NS across groups. (**I**). ATP linked OCR, NS across groups. (**J**). Proton leak OCR, NS across groups. (**K**). Maximal OCR, normoxia TS vs. IR TS, *p* = 0.0448, IR TS vs. IR TB, *p* = 0.0009. (**L**). reserve capacity OCR, normoxia TS vs. IR TS, *p* = 0.0217, IR TS vs. IR TB, *p* = 0.0097. TS, Tat-Scrambled; TB, Tat-Beclin 1. I2R4, ischemia 2 h and reperfusion 4 h; I2R6, ischemia 2 h and reperfusion 6 h; NS (not significant).

**Figure 3 cells-11-02111-f003:**
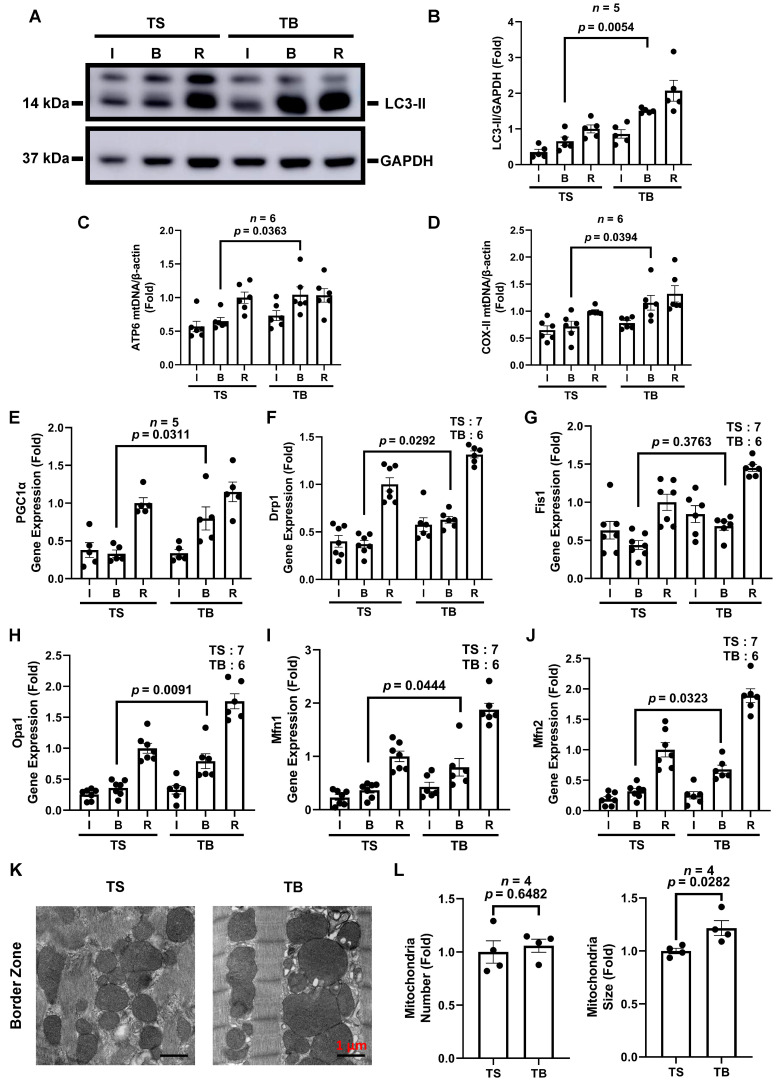
Tat-Beclin 1 treatment induces autophagy, maintains mitochondrial DNA levels, and increases PGC1a and mitochondrial dynamic gene expression and mitochondrial area in mouse heart with I/R injury. Mice were subjected to cardiac I/R injury (45 min/24 h) and treated with Tat-Scrambled or Tat-Beclin 1 during reperfusion, and then the left ventricle was cut into three distinct zones (infarct, border, and remote). Whole tissue lysate was extracted for Western blot, DNA and RNA were isolated for quantification of mtDNA, and mitochondrial biogenesis and fission and fusion related genes. EM images were also obtained. (**A**,**B**), representative Western blot, and quantification of LC3-II expression in three zones of mouse heart subjected to I/R. *n* = 5, *p* = 0.0054. (**C**,**D**), ATP6 and COX-II primers were used to determine the mtDNA copy number in three zones of I/R treated mouse heart. *n* = 6. ATP6, *p* = 0.0363. COX-II, *p* = 0.0394 (border zone). (**E**), PGC1α gene expression in the mouse heart subjected to I/R. *n* = 5, *p* = 0.0311 (border zone). (**F**,**G**), mitochondrial fission-related gene expression in mouse heart subjected to I/R. *n* = 6–7. TS vs. TB, Drp1, *p* = 0.0292. Fis1, *p* = 0.3763 (border zone). (**H**–**J**), mitochondrial fusion-related gene expression in mouse heart subjected to I/R. *n* = 6–7. TS vs. TB, Opa1, *p* = 0.0091. Mfn1, *p* = 0.0444. Mfn2, *p* = 0.0323 (border zone). (**K**,**L**), EM images of mitochondria in the mouse heart infarct border zone. Bar = 1μm. *n* = 4. TS vs. TB, mitochondria numbers, *p* = NS. Mitochondria size, *p* = 0.0282. TS, Tat-Scrambled; TB, Tat-Beclin; I, infarct zone; B, border zone; R, remote zone; NS (not significant).

**Figure 4 cells-11-02111-f004:**
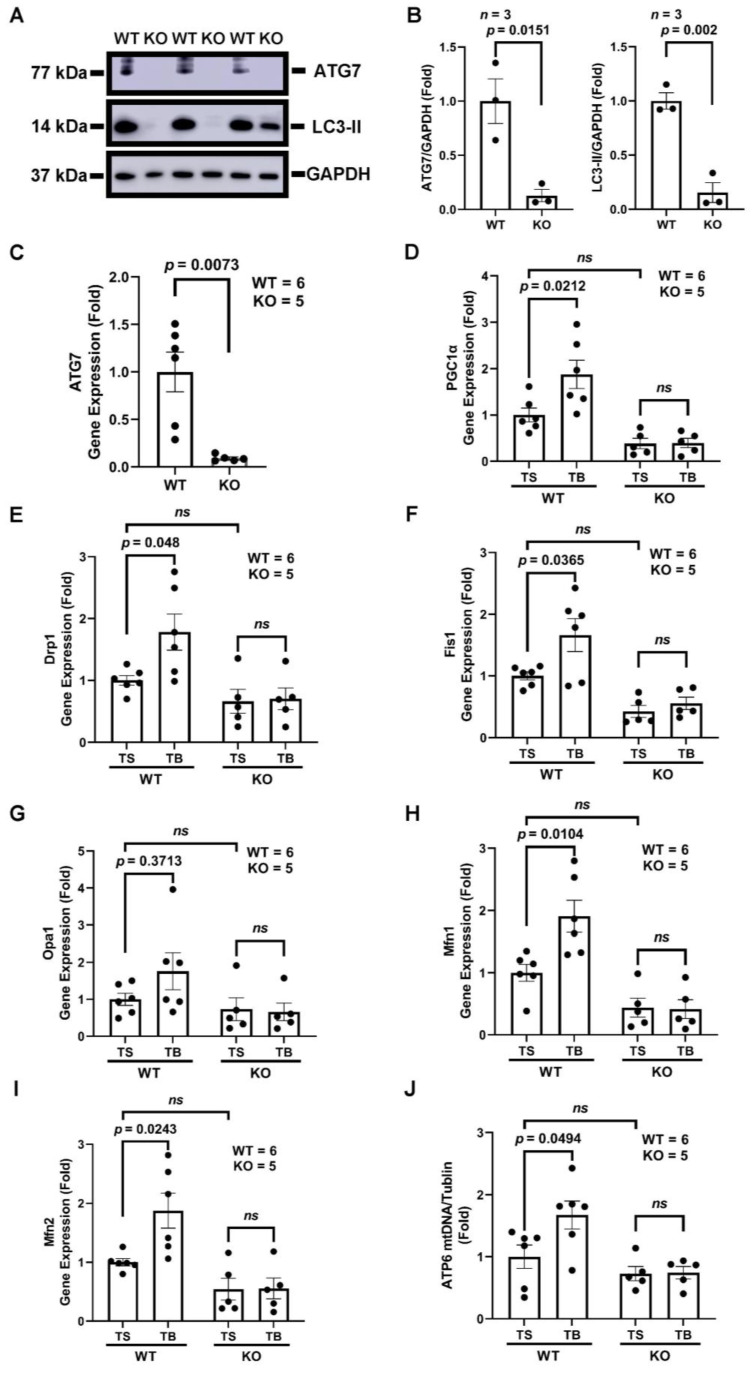
The beneficial effects of Tat-Beclin 1 on mitochondrial homeostasis in adult mouse ventricular cardiomyocytes (AMVMs) depend on autophagy. Mouse model of inducible cardiomyocyte-specific ATG7 knockout mice (ATG7 cKO (KO)) were generated. ATG7F/F treated with tamoxifen injection (WT) were used as control. AMVMs were treated with TS or TB for 2 h, DNA and RNA were extracted for qPCR and qRT-PCR, respectively. (**A**,**B**), representative Western blot, and quantification of ATG7 expression and LC3-II level in AMVMs. *n* = 3. WT vs. KO, ATG7, *p* = 0.0151. LC3-II, *p* = 0.002. (**C**), ATG7 gene expression in AMVMs. *n* = 5–6. WT vs. KO, *p* = 0.0073. (**D**), PGC1α gene expression in AMVMs. *n* = 5–6. TS vs. TB, WT, *p* = 0.0212. KO, *p* = NS. (**E**,**F**), mitochondrial dynamics fission-related gene expression in AMVMs. *n* = 5–6. TS vs. TB, Drp1, WT, *p* = 0.048. KO, *p* = NS. Fis1, WT, *p* = 0.0365. KO, *p* = NS. (**G**–**I**), mitochondrial dynamics fusion-related gene expression in AMVMs. *n* = 5–6. TS vs. TB, Opa1, WT, *p* = 0.3713. KO, *p* = NS. Mfn1, WT, *p* = 0.0104. KO, *p* = NS. Mfn2, WT, *p* = 0.0243. KO, *p* = NS. (**J**), mtDNA copy number in cardiomyocytes (AMVMs) were analyzed by qPCR of ATP6. *n* = 5–6. TS vs. TB, WT, *p* = 0.0494. KO, *p* = NS. TS, Tat-Scrambled; TB, Tat-Beclin 1; NS (not significant).

**Figure 5 cells-11-02111-f005:**
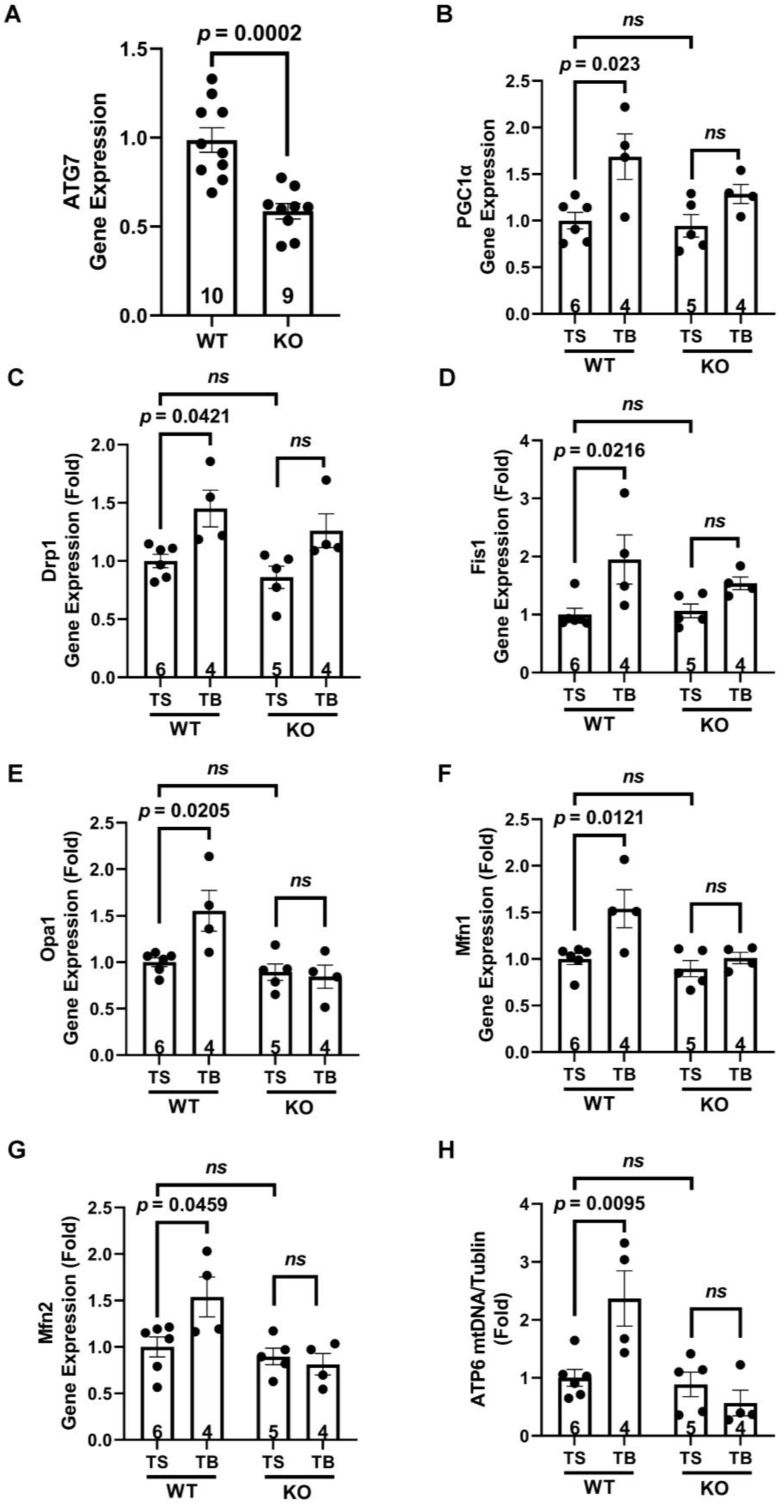
The beneficial effects of Tat-Beclin 1 on mitochondria in mouse hearts depend on autophagy. ATG7 cKO (KO) (αMHC-merCremer+; ATG7F/F injected with tamoxifen) were generated. ATG7F/F treated with tamoxifen injection (WT) were used as control. Mice were treated with Tat-Scrambled or Tat-Beclin 1 for 24 h, and the left ventricle was isolated for analysis. (**A**), ATG7 gene expression in mouse heart. *n* = 9–10. WT vs. KO, *p* = 0.0002. (**B**), PGC1α gene expression in mouse heart. *n* = 4–6. TS vs. TB, WT, *p* = 0.023. KO, *p* = NS (not significant). (**C**,**D**), mitochondrial dynamics fission-related gene expression in mouse heart. *n* = 4–6. TS vs. TB, Drp1, WT, *p* = 0.0421. KO, *p* = NS. Fis1, WT, *p* = 0.0216. KO, *p* = NS. (**E**–**G**), mitochondrial dynamics fusion-related gene expression in mouse heart. *n* = 4–6. TS vs. TB, Opa1, WT, *p* = 0.0205. KO, *p* = NS. Mfn1, WT, *p* = 0.0121. KO, *p* = NS. Mfn2, WT, *p* = 0.0459. KO, *p* = NS. (**H**), mtDNA copy numbers in mouse hearts were analyzed by ATP6 qPCR. *n* = 4–6. TS vs. TB, WT, *p* = 0.0095. KO, *p* = NS. TS, Tat-Scrambled; TB, Tat-Beclin 1; NS (not significant).

**Figure 6 cells-11-02111-f006:**
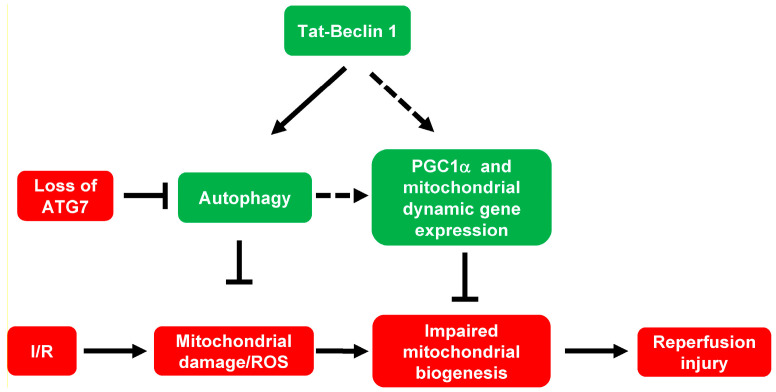
Summary of the findings. Cardiac I/R induces mitochondrial damage and ROS production. Directly activated autophagic flux by a novel and powerful autophagy-inducing peptide, Tat-Beclin 1, reduces I/R-induced ROS level and increases mitochondrial homeostasis during cardiac I/R injury. Tat-Beclin 1’s cardioprotective effects are dependent on the crucial autophagy-related gene, ATG7. Induction of autophagic flux also seems to induce mitochondrial biogenesis mediated by PGC1α and possibly increase mitochondrial dynamics (fission and fusion). The mechanisms of autophagic flux-induced mitochondrial biogenesis and mitochondrial dynamics need further investigation.

## Data Availability

Data is contained within the article or Appendix A. Detailed methods can be found in online Expanded Materials and Methods.

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
