# Peer review of "Activation of Autophagic Flux Maintains Mitochondrial Homeostasis during Cardiac Ischemia/Reperfusion Injury"

_cells, 2022, doi:10.3390/cells11132111_

Round 1

Reviewer 1 Report

Thanks for the point-by-point reply.

Author Response

Thanks for the review and suggestions. We have improved the writing and have the manuscript professionally proofread and fixed spelling and grammar errors.

Reviewer 2 Report

The revised manuscript is much improved. The additional experiments investigating how TB reduces IF-induced changes in mitochondrial ROS, oxidative metabolism, morphology, ATP and cell death have greatly supported the authors' hypothesis compared to the original submission. My other comments have been adequately addressed.

Author Response

Thanks for the initial comments. They guided our experiments to make the manuscript better. We have the manuscript professionally proofread and fixed spelling and grammar errors.

Reviewer 3 Report

the authors have addressed my concerns

Author Response

We thank the reviewers for the comments and suggestions. 

We also have the manuscript professionally proofread and fixed spelling and grammar errors.

This manuscript is a resubmission of an earlier submission. The following is a list of the peer review reports and author responses from that submission.

Round 1

Reviewer 1 Report

He et al. show the use of a Beclin-Tat peptide (TB) to activate the mitochondrial flux during ischemia/reperfusion (IR). Authors have tested this specific autophagy-inducing peptide in vitro and in vivo in a mouse cardiac I/R model. This approach have been integrated into the need for efficient treatments of acute myocardial infarction.

Major issues:

  1. The differences that authors observe are poor. Although statistically significant in some occasions, it is difficult to believe that it correlates with any biological significant effect. In figure 1 the effect of TB peptide it is difficult to observed. This figure contrast with the effect observed in the originally described figure where this TB peptide was characterized (doi: 1038/nature11866) .

Figure 1 in this manuscript.

Original paper

This fact suggest that the TB peptide that authors are using have not much effect on the cardiac context. Consequently it raises concerns about the biological significance of the results.

  1. The difference observed in mitochondrial markers are very difficult to observe. The authors should show other ways to show mitochondrial fitness like electron microscopy in cardiac tissues (in vivo) or oxygen consumption experiments (SeaHorse like assays) in vitro.

  1. If TB can have a beneficial effect on I/R protection authors should show an echocardiography to demonstrate the TB benefit, or at least they should properly show a quantification of area at risk, infarct size using a recognized staining protocol, see for example DOI: 10.1038/ncomms14780. Authors should observe a significant change in infarcted area using TB.

Reviewer 2 Report

He et al. investigates mitochondrial homeostasis in a model of cardiac ischemia/reperfusion.

The topic is of much interest as underlying mechanisms are not fully understood and effective therapy options needs to be developed.

Overall, the manuscript is well written. However, some issues needs to be addressed and some additional results may provide the findings with more consistency.

Comments:

Materials & Methods:

  1. Please specify the sex of the mice and rats used.
  2. Where the NRVM passaged before used in the simulated I/R experiment? If yes, which passages of cells were used?
  3. Please indicate the concentration of the fluorescence dyes used for the experiments.

Results/Supplement:

  1. For Figure 1A, 3A, 5A is not a full gel given in the supplement (it is cut for 14 kDa and 37 kDa).
  2. Scale bar is not given for all fluorescence images in Figures and Supplement. As scale bar seems to differ, please include for all images.
  3. As H2DCFDA is not specific for mitochondrial ROS production, it might be helpful to include data for mito-ROS specific fluorescence dyes.
  4. Beside intact/total ratio did you measure other markers of mtDNA damage?
  5. What is the effect of treatment on the oxygen consumption rate?
  6. Did TB treatment effects mitochondrial morphology e.g., measured by electronic microscopy?

Reviewer 3 Report

The authors of this manuscript recently published findings that cardiac dysfunction from ischemia/reperfusion (I/R) injury could be reduced through administration of a Tat-Beclin-1 (TB) peptide (Xie et al., 2021). TB blocks inhibition of Belclin-1, releasing it to activate autophagy. TB was then reported to reduce markers of I/R-induced autophagy in the heart via an Atg7-dependent mechanism. Here, He et al. seek to understand whether the beneficial effects of TB in I/R involve mitochondrial biogenesis and homeostasis as mitochondrial dysfunction is known to cause increased ROS damage, alter metabolic substrate utilization, and cell death from I/R injury. Both a simulated I/R in vitro model using neonatal rat and adult mouse cardiomyocyte and an in vivo I/R model in wildtype and Atg7 knockout animals was used. Readouts include autophagic conversion of LC3-I into LC3-II, fluorescent detection of ROS levels and mitochondrial membrane potential, mitochondrial DNA abundance, and a survey of PGC1-alpha and mitochondrial fission/fusion gene expression. The purpose of the study was to test the effects of autophagic flux induced by TB on cardiomyocyte mitochondrial biogenesis and homeostasis during I/R injury. While the findings in this paper suggest these processes may occur with TB, the data is not sufficiently developed and critical experiments are missing to demonstrate that mitochondrial biogenesis and homeostasis is affected.

Major points:

  1. There are many instances throughout the manuscript in which data interpretation is not consistent with the data presented. For example, there is consistent use of “increase” when comparing TB vs. TS under I/R conditions. However, these data show that I/R treatment decreases the measured parameter with TS relative to normoxic conditions. TB then restores those levels in I/R to a level comparable to normoxic conditions, without a further increase. It may therefore be better to change any quantitative statements to “preserves” or “maintains”, etc. as is sometimes written in other sentences.
  2. Figure 1C, D. The image panels are of different magnifications. Might this affect image quantification? Please describe methods used for quantification in greater detail. How many cells or fields were quantified in each of the N=3 independent experiments?
  3. Figure 2E, F. The upper right panel appears to be out of focus. This would likely reduce the accuracy of TMRM quantification. Was image quantification performed cell-by-cell or field-by-field? How many cells or fields were quantified in each of the N=3 independent experiments?
  4. Figures 3&4. I/R is known to alter processes even in the remote region. Does peptide administration alone alter LC3-II, mtDNA, or RNA levels in unstressed hearts compared to I/R remote tissue?
  5. Conclusions regarding mitochondrial autophagy require more definitive assays. The widely varying and small n=3 sample dataset for WT + TB in Figure 6B-H could be improved with increased sample numbers to strengthen whether the increase in gene expression is indeed significant. There is an insignificant tendency for an increase in KO animals where there are more samples and less variance. A more direct approach could address whether autophagosome/lysosome membranes or markers are evident around mitochondria and decrease with TB.
  6. Conclusions regarding mitochondrial fission or fusion also require more definitive assays. The data cannot properly interpret what is happening to fission or fusion based on RNA levels alone as changes in PGC1-alpha and fission/fusion gene RNAs may not lead to altered protein levels, and confusingly, both fission and fusion levels are increased with I/R. Quantitative measurements of mitochondrial size and indicators of fission vs. fusion (e.g. phospho-Drp1) would add important support to this idea.
  7. Similarly, an improved approach is needed if mitochondrial homeostasis vs. dysfunction is to be understood. A correlation with total cellular ROS levels and mitochondrial membrane potential is not quite sufficient in itself. Does the integrity of individual mitochondria improve with TB? Are there mitochondrial functions and activities that are altered during I/R and normalized with TB? There are several references to the effects of mitochondrial dysfunction that are hypothesized to occur but are not tested (e.g. cell death, respiratory complex function, glycolytic shifting, etc.). Some experiments testing these outcomes is expected to improve the significance of the manuscript.
  8. What is the mechanism for TB and Atg7 to induce the nuclear expression of mitochondrial genes? It is difficult to envision that the induction of PGC1-alpha RNA is driving potential changes other mitochondrial genes within the 2 hours I/R experiment shown in Figure 5. The authors indicate that autophagy induced PGC1-alpha biogenesis is unknown (lines 512-514). However, might NRF1 (NFE2L2), which is known to induce PGC1-alpha, serve as that missing link and found to be operative in this I/R model (e.g. PMID 26391655)?
  9. Line 480-481. No data is presented for clearance of damaged mitochondria.

Minor points:

  1. Line 94. Ref 14 appears to be incorrect. Kamatsu, et al. 2005 (PMID 1586687) may be more appropriate for citing the ATG7-f/f mouse. This paper is cited in the Supplement.
  2. Please also check whether the genotyping primers in the Supplement detect floxed versus knockout alleles. Are floxed PCR products expected to cause a difference in 1000 bp or give no band as written?
  3. Line 340. “medicated” should be “mediated”?
  4. Line 391. “relative” should be “related”?
  5. Line 472 has an incomplete sentence.